# Biological Mineralization of Methyl Orange by *Pseudomonas aeruginosa*

**Asad Ullah Khan [1], Muhammad Zahoor [2],[*], Mujaddad Ur Rehman [1], Abdul Bari Shah [3], Ivar Zekker [4],[*], Farhat Ali Khan [5], Riaz Ullah [6], Ghadeer M. Albadrani [7], Roula Bayram [8],[9] and Hanan R. H. Mohamed [10]**

[1] Department of Microbiology, Abbottabad University of Science and Technology, Havelian, Abbottabad 22010, KPK, Pakistan; asad.microbiologist@gmail.com (A.U.K.); mujaddad@aust.edu.pk (M.U.R.)
[2] Department of Biochemistry, University of Malakand, Chakdara, Dir Lower 18800, KPK, Pakistan
[3] Division of Applied Life Science (BK21 Plus), Institute of Agriculture and Life Sceience, Gyeongsang National University, Jinju 52828, Korea; abs.uom28@gmail.com
[4] Institute of Chemistry, University of Tartu, 14a Ravila St., 50411 Tartu, Estonia
[5] Department of Pharmacy, Shaheed Benazir Bhutto University, Sheringal, Dir Upper 18050, KPK, Pakistan; farhatkhan2k9@yahoo.com
[6] Department of Pharmacognosy, College of Pharmacy, King Saud University, Riyadh 11451, Saudi Arabia; rullah@ksu.edu.sa
[7] Department of Biology, College of Science, Princess Nourah bint Abdulrahman University, B.O. Box 84428, Riyadh 11671, Saudi Arabia; gmalbadrani@pnu.edu.sa
[8] Pharmacy Program, Department of pharmaceutical science, Batterjee Medical College, Jeddah 21442, Saudi Arabia; lab3.jed@bmc.edu.sa
[9] Department of Analytical Chemistry, Faculty of Pharmacy, Aleppo University, Aleppo 15310, Syria
[10] Zoology Department, Faculty of Science, Cairo University, Giza 12613, Egypt; hananeeyra@cu.edu.eg
[*] Correspondence: mohammadzahoorus@yahoo.com (M.Z.); ivar.zekker@ut.ee (I.Z.)

**Abstract:** Due to its recalcitrant and carcinogenic nature, the presence of methyl orange (MO) in the environment is a serious threat to human and animal life and is also toxic to plants. MO being recalcitrant cannot be effectively reclaimed from industrial effluents through physical and chemical approaches. Biological methods on the other hand have the potential to degrade such dyes because of their compatibility with nature and low chances of adverse effects on the environment. Bacteria, due to their fast growth rate and capability of surviving in extreme environments can effectively be used for this purpose. In the current research study, *Pseudomonas aeruginosa* was isolated and characterized using 16rRNA from textile wastewater. In the preliminary tests it was found that *Pseudomonas aeruginosa* has the ability to degrade and mineralize methyl orange effectively. The physicochemical conditions were then optimized, in order to get maximum degradation of MO which was achieved at 37 °C, a pH of 7, a low salt concentration of 0.1 g/15 mL, a high carbon source of 0.6 g/15 mL, and 72 h experimental time. In a single set of experiments where all these optimum conditions were combined, 88.23% decolorization of the selected dye was achieved. At the end of the experimental cycle, the aliquots were homogenized and filtered. The filtrates were subjected to FTIR and GC-MS analysis where azo linkage breaking was confirmed from the FTIR spectra. The filtrates were then extracted with ethyl acetate and then passed through a silica gel column. On the basis of Rf value (TLC plates used) similar fraction were combined which were then subjected to NMR analysis. The compounds detected through GC-MS, peaks were not observed in proton and C-13 NMR. Instead, solvent and some impurity peaks were present, showing that complete mineralization of the dye had occurred due to the action of different bacterial enzymes such as azoreductase, peroxidases, and classes on MO. The prosed mechanism of complete mineralization is based on spectral data that needs to be verified by trapping the individual step products through the use of appropriate inhibitors of individual enzymes.

**Keywords:** bacterial degradation; methyl orange; *Pseudomonas aeruginosa*; wastewater

## 1. Introduction

Azo dyes are broadly utilized for dyeing paper, textile fibers, and leather. They are also used in pharmaceutical, cosmetic, and some other industries. As compared to natural dyes, azo dyes are more stable to temperature, chemicals, and detergents. They are low cost, easily available, bright in color and can be easily degraded by microbes. These characteristics have considerably increased their use worldwide [1]. Thousands of tons of azo dyes are used in the textile industry annually where more than 29,000 tons in the unaltered form are being discharged into the water and soil each year. Almost 70% of azo dyes are utilized in the textile industry globally [2]. Methyl orange is a synthetic anionic azo dye that is organic, sulfonated, heterocyclic, and has high water solubility. Methyl orange dye is widely utilized in a variety of industries, including textiles, dyeing, leather, and paper. In many research laboratories, methyl orange is used as a pH indicator [3].

The direct discharge of methyl orange into water sources and soils without proper treatment is dangerous for terrestrial and aquatic organisms as well as humans and other animals. Even at a low concentration methyl orange interrupts light penetration into deep water thereby adversely affecting photosynthesis. It also effects gaseous solubility in water, required for animals living in the water [4]. The refractory dye methyl orange causes intestinal cancer, lung cancer, hypersensitivity, allergy, and dermatitis in living organisms including human beings. Its high toxicity, teratogenicity, carcinogenicity, and mutagenicity have already been confirmed in humans and animals [5]. It also has the additional effect of decreasing crop productivity, soil fertility, and biodiversity of plants by increasing salt in the soil. Therefore, methyl orange dye needs proper treatment before being discharged into water and soil [6].

Many methods (physical, chemical, and biological) are used for the degradation and mineralization of methyl orange. It should be noted that the physical and chemical procedures are only useful for the decolorization of dyes and are not applicable for the degradation of dyes, and they produce extra sludge that needs further treatment. On the other hand, the use of biological methods is eco-friendly, cost-effective, and does not produce sludge [7]. In biological methods, the most preferably employed organisms are bacteria that have a rapid growth cycle and thus produce high amounts of biomass needed for the required degradation process. Apart from this, bacteria have the highest potential to degrade azo dyes when compared to fungi, algae, and plants as in spite of a high replication rate, bacteria can tolerate tough conditions in varied environments. Bacteria have the ability to mineralize or degrade azo dyes into water, carbon dioxide and some partially degraded products [8]. Many bacterial species have been reported which have the ability to degrade and mineralize azo dyes, such as *Shewanella species*, *Bacillus species*, *Serratia species*, *Thermus species*, *Pseudomonas species*, *Aeromonas species*, and *Serratia species* [9]. Bacteria secrete enzymes such as azoreductase, peroxidases, and lasses, which can degrade and mineralize azo dyes. Research studies have been conducted on degradation and mineralization of methyl orange (MO) by bacteria [10]. According to Haque et al. [11] *Aeromonas hydrophila* has the ability to degrade methyl orange up to 90%. Yang et al. [12] reported that *Stenotrophomonas acidaminiphila* degrades the dye methyl orange up to 95%. According to Al-Ansari et al. [13] the *Enterobacter aerogenes* ES014 has the ability to degrade methyl orange up to 75%. Masarbo et al. [14] also reported that *Kerstersia* sp. have the ability to degrade and mineralize methyl orange up to 96%. Although the extent of decolorization of MO has been reported in the cited studies, none of the research has given attention to the nature of the metabolites formed. Such types of studies are important from the environmental point of view, to know whether the resulting metabolites are toxic or eco-friendly.

Due to these facts and the wide range of applications of MO across several industries and its potential to have major consequences for the environment and living organisms, the present study was designed, firstly to degrade the dye effectively by *Pseudomonas aeruginosa* and then to isolate and characterize the metabolites formed during the decolorization process as the significance and nature of the degradation products produced by microor-

ganisms have not been adequately studied previously. Out of the tested bacterial strains, *Pseudomonas aeruginosa* had the highest ability to degrade methyl orange to a great extent as compared to other bacteria tested. *Pseudomonas aeruginosa* is a facultative bacterium that can grow both aerobically and anaerobically, and has the ability to rapidly grow in azo dye contaminated water. For the characterization of methyl orange metabolites, no study has been conducted previous to this research. Additionally, the effect of many physicochemical factors on dye degradation were explored in order to determine the circumstances that result in a high degradation and mineralization of the dye. After complete degradation of the dye under optimum conditions, the metabolites were separated using column chromatography. For the characterization of metabolites, different techniques were used, such as FTIR, GC-MS, and NMR.

## 2. Materials and Methods

### 2.1. Chemicals

MO and other chemicals required for the experimental work were of the highest purity analytical grade and were used without any additional purification. The chemicals such as; sodium chloride, nutrient broth, hydrochloric acid, sodium hydroxide, and glucose were purchased from Sigma Aldrich, Germany.

Methyl orange has a molecular weight of 327.33 g/mol and its molecular formula is $C_{14}H_{14}N_3NaO_3S$. Its chemical structure is given below in Figure 1 [15].

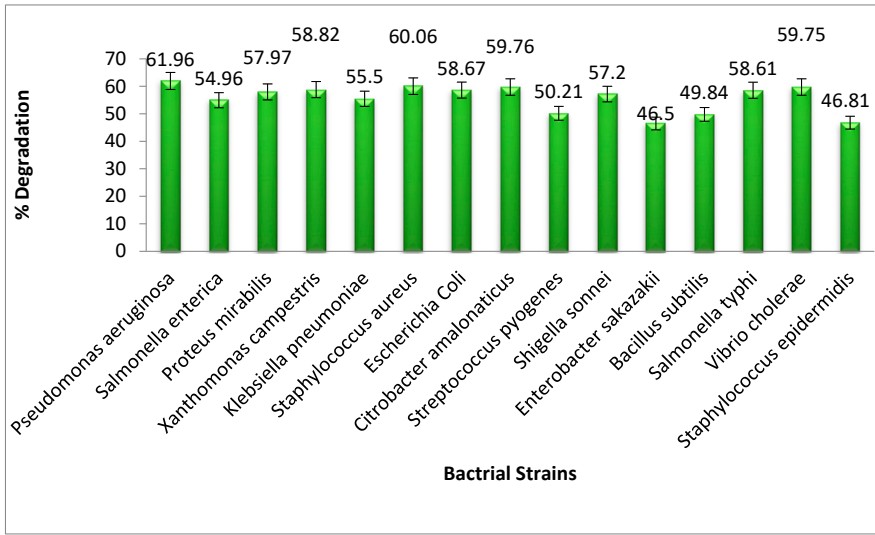

**Figure 1.** Methyl orange dye chemical structure.

Initially, 15 strains of bacteria were screened for decolorization capability of MO. Their details are shown in Figure 2. All these strains have been isolated from industrial effluents. The experimental conditions were; 20 ppm of dye solution, a temperature of 37 °C, and 15 mL of broth medium combined with bacterial inoculum; a suspension containing $10^8$–$10^9$ CFU/mL and cultured for three days at neutral pH.

**Figure 2.** Different bacterial strains were used for the degradation of methyl orange (at neutral pH, 37 °C, 20 ppm of dye, 15 mL of nutrient broth, and an inoculum of $10^8$–$10^9$ CFU/mL of bacteria were cultured for three days).

The MO degradation experiments on *Pseudomonas aeruginosa* were performed in test tubes containing 15 mL broth agar and 5 mL of 20 ppm solution of dye. First, the nutrient broth containing the methyl orange was autoclaved for 15 min at 121 °C to kill unwanted microorganisms if present. The tubes were then incubated using an incubator for 3 days at 37 °C. After incubation, the tubes were subjected to centrifugation for 20 min at 5000 rpm to separate the pellet and supernatant. The upper liquid portion was then evaluated for the remaining MO concentration using a visible spectrometer at 464 nm [2].

The following formula was used to estimate the extent of degradation:

$$\% \ Decolorization = \frac{Initial \ absorbance - final \ absorbance}{Initial \ absorbance} \times 100\% \tag{1}$$

As mentioned before, *Pseudomonas aeruginosa* was the most effective strain that mineralized methyl orange among the other strains of bacteria; hence it was employed in further decolorization investigations. Prior to experiments the confirmation of the bacterial strain was performed as described below.

### 2.2. Molecular Identification of Bacterial Isolates

### 2.2.1. Extraction of Bacterial DNA

DNA of the selected bacterial strain was extracted from a pure culture by incubating it for 24 h at 37 °C. Then a single colony of bacteria was suspended in 250 μL of sterile water, which was nuclease free. The bacterial cells were lysed by heating them at 100 °C for 16 min and centrifuged at 10,000 rpm for 3 min to remove all the cell debris. For the DNA template, 5 μL of cell lysate was used [16].

### 2.2.2. PCR Amplification

A GenePro Thermal Cycler was used for PCR amplification of DNA. The 20 μL of PCR master mixture contained 1 μL of 16S rRNA reverse primer (5′-CACTGGTGTTCCTTCCT ATA-3′), 1 μL of 16S rRNA forward primer (5′-.GACGGGTGAGTAATGCCTA.-3′), 3 μL of nuclease free water, 5 μL of DNA template and 10 μL of master mix. The thermo cycler was programmed to incubate the samples at 95 °C for 5 min, then it was run for 35 cycles at 94 °C for 15 s, 53 °C for 30 s, and 72 °C for 45 s, with a final extension of 10 min at 72 °C, before stopping the process. The amplification size of the gene DNA was 617 base pairs. The PCR amplified product was loaded onto a 2 percent agarose gel for electrophoresis [17] along with a 600-bp DNA marker (Fermentas Life Sciences, New York, NY, USA), stained with ethidium bromide (10 g/mL). A UV-trans illuminator was used to examine the PCR products, and the remaining PCR products were stored for further sequencing. The 16S rRNA sequences were analyzed using an ABI genetic analyzer. The nucleotide sequences of the 16S rRNA gene were compared using the online BLAST search engine [18]. The isolated bacteria showed a 100% similarity with the bacterial strain *Pseudomonas aeruginosa*.

### 2.3. Physicochemical Parameter Optimization for Dye Degradation

### 2.3.1. Dye Concentration Effect

The *Pseudomonas aeruginosa* bacterial strain was used to degrade methyl orange for three days at five different dye concentrations (100, 80, 60, 40, and 20 ppm). In each experiment, 15 mL broth spiked with varied concentrations of dye in 5 mL were inoculated with the selected bacterial strain. After 3 days of incubation, each tube was centrifuged for 20 min at 5000 rpm to separate the pellet and supernatant. The concentration of remaining dye in the supernatant was used to estimate the percent degradation using a visible spectrometer at 464 nm [15].

### 2.3.2. The Effect of pH on Dye Degradation

To investigate the degradation or decolorization of methyl orange and the effect of pH, thirteen test tubes containing 20 ppm dye concentration and nutrient broth were incubated at 37 °C for three days with *Pseudomonas aeruginosa* at various pH levels ranging from 1

to 13. About 1 N NaOH and 1 N HCl were used for adjusting pH values of the solutions. Test tubes were inoculated with bacteria after the adjustment of the pH of each. After 3 days of incubation, each sample in the test tube was centrifuged for 20 min at 5000 rpm to separate the pellet and supernatant from each other. The supernatant was used in the determination of percent degradation of remaining dye molecules through absorbance using a visible spectrometer [19].

### 2.3.3. Temperature Effect on Dye Degradation

Methyl orange solutions (20 ppm) and *Pseudomonas aeruginosa* were incubated at different temperatures of 50, 45, 37 and 25 °C. After three days of incubation at various temperatures, each culture tube was evaluated for the remaining dye concentration as mentioned above.

### 2.3.4. Effects of Glucose on Dye Degradation

To enhance the degradation process, an extra carbon source was required. The nutrient broth in test tubes containing 20 ppm methyl orange was supplemented with different concentrations; 0.1, 0.2, 0.3, 0.4, 0.5, 0.6, 0.7, and 0.8 g of glucose and incubated for three days at 37 °C. After three days of incubation at different glucose concentrations, each culture tube was measured with a UV-Visible spectrophotometer in the same way as described above.

### 2.3.5. Effect of NaCl (Sodium Chloride) on Dye Degradation

NaCl (Sodium chloride) is one of the main components in seawater and can affect the dye degradation process by bacteria. To study the effect of salt concentration, seven test tubes were prepared containing different concentrations; 0.1, 0.2, 0.3, 0.4, 0.5, 0.6, and 0.7 g of NaCl and incubated at 37 °C for three days. Formula 1 was used to figure out the percent degradation of dye after three days of incubation.

### 2.3.6. Dye Degradation and the Impact of Time

Time has a great impact on dye degradation as the bacterial growth rate is time dependent. If the bacterial growth is high and enough time is given, large amounts of biomass will be produced which in turn will warrant high rates of degradation. OECD guidelines were followed. According to these guidelines the standardized time for degradation permissible is 28 days whereas *P. aeruginosa* is capable of growing in conditions of extremely low nutrient content. This species can survive and proliferate in water for up to 100 days or longer. In order to investigate the influence of time on dye degradation, a 250 mL flask containing 20 ppm of methyl orange dye and nutrient broth was incubated for 25 days at 37 °C. Every day, after each 24 h of incubation, 10 mL samples were taken and centrifuged for 20 min at 5000 rpm to separate the supernatant and pellet. The supernatant was used for the determination of percent degradation using a UV/visible spectrometer.

### 2.4. Degradation of Methyl Orange Dye under Optimal Conditions

A single experiment was conducted to test the synergistic effect of the optimal conditions (dye concentration, optimal pH, temperature, NaCl concentration, glucose concentration, and time) established in the previous experiments to get maximum degradation. The concentration of the remaining dye in solution was determined as described above.

### 2.5. Degradation Metabolites Extraction, Isolation, and Identification

The treated samples under optimum conditions were crushed and centrifuged for 20 min at 5000 rpm at room temperature. After centrifugation, the pellet and supernatant were separated. To extract the dye metabolites, the supernatant was used. The supernatant was extracted with ethyl acetate by shaking vigorously for 30 min. The ethyl acetate was evaporated at 40 °C in order to obtain a solid extract of dye metabolites. After extraction of solid metabolites, some portions were again dissolved in ethyl acetate, and filtered with filter paper. The remaining part was subjected to size-based metabolite purification using

column chromatography. The solid extract of metabolites was mixed with silica gel mesh size 70–230 to make a slurry and left for some time to dry. It was necessary to pack silica gel into a glass column with a 4.5 cm diameter and an 80 cm height and fill it with a 45 cm height of silica gel before washing it with n-hexane for about 600 mL before loading the sample. The slurry, which was air dried, was loaded onto the glass column by washing with n-hexane and then elution with different ratios of an n-hexane and ethyl acetate (5:1, 2:1, 1:1, 1:5, 1:2) solvent system. The different ratios of eluents were collected from the column and, from the TLC (Thin-Layer Chromatography) profiling. Similar fractions of metabolites were combined for re-chromatography. During column chromatography, 5 mL fractions were collected during the elution of 80 mL of each of the solvents. Thin-layer chromatography was used for single metabolite extraction confirmation. Each of the extractions was further subjected to analysis by NMR to confirm the structure of metabolites.

### 2.5.1. GC-MS Analysis

An Agilent USB-393752 (Agilent Technologies, Palo Alto, CA, USA) gas chromatograph was used to identify the metabolites in the sample mixture through a siloxane capillary column (30 m × 0.25 mm × 0.25 μm film thickness; Restek, Bellefonte, PA, USA) HHP-5MS 5% phenyl methyl. It was equipped with an FID detector for identifying metabolites. For one minute, the oven temperature was kept at 70 °C before being raised to 180 °C in five minutes. The temperature was then increased to 280 °C for 20 min to complete the research experiment. The injector temperature was 220 °C, but the detector temperature was 290 °C. The carrier gas was helium, with a flow rate of one milliliter per minute. In a split-less mode, approximately 1 mL of sample was manually injected.

The metabolites formed were characterized using GC/MS. An Agilent HP-5973 type machine was used for this purpose equipped with a mass selective detector in the electron impact mode (Ionization energy: 70 eV), which functioned under the same procedures as the GC analysis. The spectra of extracted metabolites were compared in terms of their retention time with those reported in the scientific literature (Wiley and NIST libraries).

### 2.5.2. Metabolite Analysis Using FTIR and H-NMR

FTIR analysis was performed using a Perkin Elmer Spectrum (Model No: 103385) before and after methyl orange dye degradation by *Pseudomonas aeruginosa*. The dye metabolites, which were isolated and purified through column chromatography, were further used for NMR (Nuclear Magnetic Resonance) analysis.

## 3. Results and Discussion

### *3.1. The Efficient Bacterial Strains among the Tested Strains*

Different bacterial strains have the ability to degrade the azo dyes to a great extent. In the current research study, 15 different bacterial strains were tested for methyl orange dye degradation, in which *Pseudomonas aeruginosa* was found to be the most effective for the removal of dye from the water sample. As compared to other bacteria, *Pseudomonas aeruginosa* had the ability to degrade the dye up to 61.96%. Figure 2 shows the overall degrading capability of all the strains that were tested.

### *3.2. Dye Degradability and Dye Concentration*

Figure 3 shows the relation between the dye concentration in ppm and the dye degradation ability of *Pseudomonas aeruginosa*. The highest degradation of methyl orange was recorded at a 20 ppm solution of the dye. As the concentration of dye increases from 20 ppm, the bacterial degradation of dye decreases because the high concentration of dye is toxic to bacterial growth. Because of this, low concentrations of dye should be utilized whenever possible while researching the degradation of dyes by *Pseudomonas aeruginosa*. Khan et al. [2] studied this by decreasing the concentration of dye which could increase the degradation of dye by bacteria. Additionally, it is possible that the methyl orange metabolites and high concentration of dyes interact with the active site of azoreductase,

which is responsible for the degradation of dyes. As a result, microorganisms may be able to break down more dye at lower concentrations than at high concentrations.

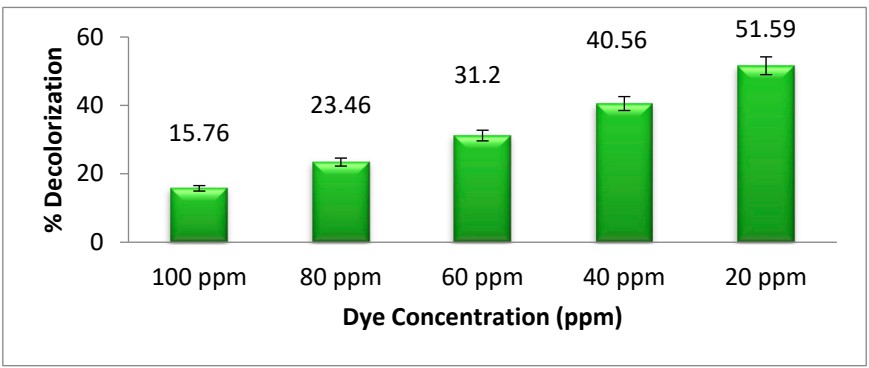

**Figure 3.** Effect of dye concentration on methyl orange degradation.

### 3.3. The Effect of pH on Methyl Orange Degradation

The effect of pH on methyl orange degradation by *Pseudomonas aeruginosa* is shown in Figure 4. The highest degradation of dye was observed at neutral pH. When the pH of a solution changes from its neutral state, the efficacy of dye degradation by microorganisms is decreased. This is because, below and above the neutral pH, bacterial growth degreases. Previous investigations into azo dye degradation at neutral pH confirm the findings of our study [20]. According to Velusamy et al. [21], the degradation and decolorization of methyl orange by bacteria is achieved at neutral pH. Our research results show that the highest degradation of methyl orange by *Pseudomonas aeruginosa* was 60.01% at neutral pH. When the pH of the bacterial medium increases or decreases from the neutral pH, the bacterial growth and enzyme activities decrease gradually.

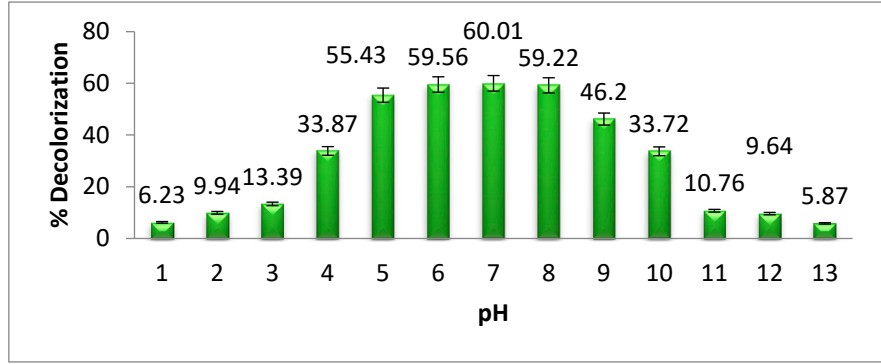

**Figure 4.** Effect of pH on methyl orange degradation.

### 3.4. Temperature Effect on Dye Degradation

The optimum temperature is very important for bacterial growth. Figure 5 shows the effect of different temperatures on bacterial degradation of methyl orange. The maximum degradation or decolorization of azo dye was achieved at 38 °C, which is about 59.32%.The biodegradation of dye is decreased above and below the optimum temperature due to the decrease in bacterial growth. According to Khan et al. [2], the optimum temperature for *Pseudomonas aeruginosa* azo dye degradation is 38 °C, which supports our research results. As the temperature increases or decreases from the optimum temperature, the bacterial growth and azoreductase activity decrease [22].

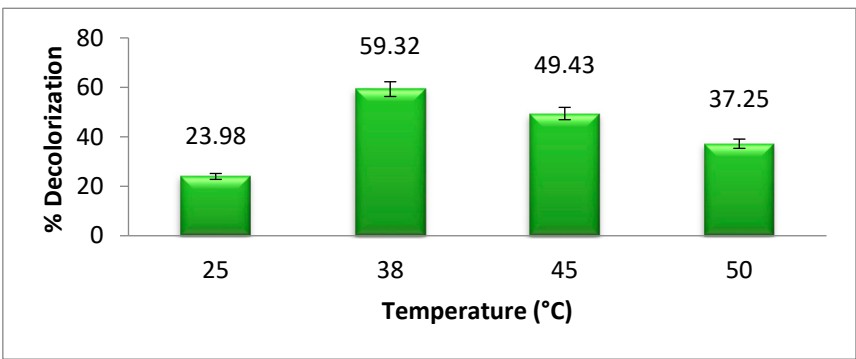

**Figure 5.** Effect of temperature on methyl orange degradation.

### 3.5. Effect of Carbon Supplementation on Methyl Orange Degradation

Glucose is a basic and important source of carbon for microorganisms. In the presence of carbon supplements, the bacterial biomass increases. As a result, the biodegradation of dye also increases. In azo dyes, some dyes are complex in their structure, which requires a large amount of bacteria to degrade the dye. Because of this, the carbon source increased the rate of bacterial growth [23]. In Figure 6, we show the *Pseudomonas aeruginosa* degradation of methyl orange at different concentrations of glucose supplements, which act as a carbon source for bacteria. Our results show that as the glucose concentration increases from 0.1 to 0.6 g per 15 mL, the dye degradation increases gradually. At a concentration of 0.6 g per 15 mL, selected bacteria may degrade up to 65.82% of the particular dye. A rise in glucose content from 0.6 g per 15 mL leads to a gradual decrease in the degradation of dyes. This is because high concentrations of glucose have a negative effect on bacterial metabolism, which is called "sugar catabolic suppression". This happens when glucose is present in high quantities [24].

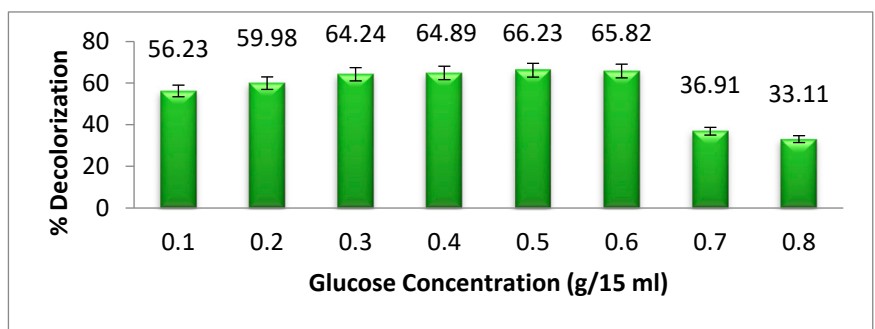

**Figure 6.** Degradation of methyl orange at different glucose concentration.

### 3.6. Effect of Sodium Chloride (NaCl) Concentration on Dye Degradation

In light of the fact that industrial effluent comprises a high concentration of salts (NaCl), we investigated the selected dye degradation by bacteria in the presence of a variety of salt concentrations [25]. The highest dye degradation of methyl orange was found at 0.1 g per 15 mL of NaCl supplementation (Figure 7). The dye degradation decreases as the salt concentration increases from 0.1 g per 15 mL. A high concentration of salt promotes plasmolysis of bacterial cells, which results in a reduction in the development of bacteria and, as a consequence, a reduction in the breakdown of dye [26].

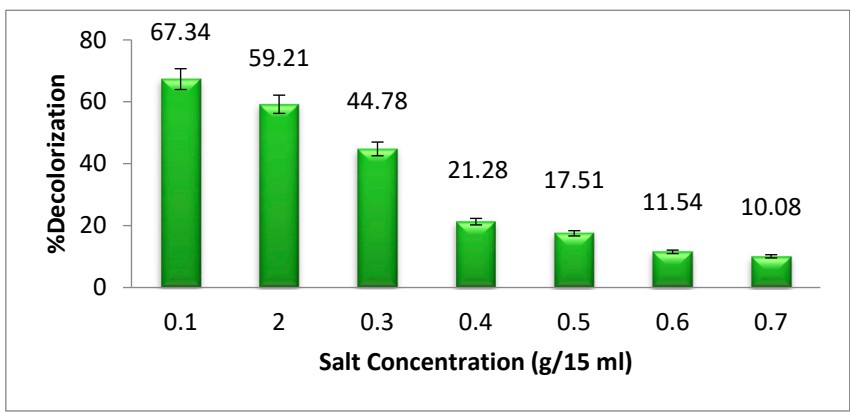

**Figure 7.** Effect of salt concentration on methyl orange degradation.

### 3.7. The Effects of Time

The effect of time on the breakdown of methyl orange by *Pseudomonas aeruginosa* is shown in Figure 8. It has been shown that, as time passes, the degradation of selected dyes becomes more significant at a specific interval of time. Although, after 3 days, there will be little change in the degradation of dye. Therefore, the optimum time for methyl orange degradation was noted as 3 days. The decolorization of methyl orange increased with time, up to 3 days. After 3 days of incubation, there was no significant change in dye degradation by bacteria. Initially, the solution contained large amounts of nutrients for bacterial growth but as time passes the growth decreases, due to an increase in biomass and a decrease in nutrients [27].

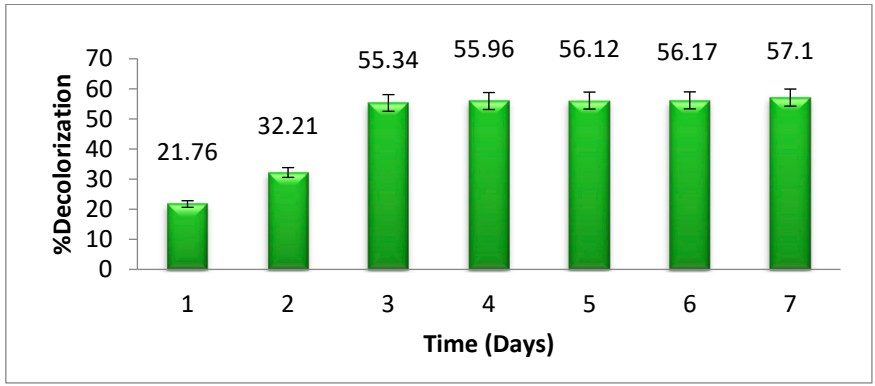

**Figure 8.** Effect of time in days on % degradation of methyl orange.

### 3.8. Degradation of Methyl Orange Dye at Optimum Physiochemical Conditions

In this study, the physiochemical parameters of methyl orange degradation or decolorization by selected bacteria, *Pseudomonas aeruginosa*, were determined, including the temperature, dye concentration, salt concentration, glucose concentration, pH of the solution, and duration (in days). All the optimum physiological conditions were applied in a single experiment. At optimal physiochemical conditions, the degradation of the azo dye (methyl orange) reached 88.23%.

### 3.9. FT-IR Analysis of Methyl Orange

The Fourier-transform infrared analysis spectra of untreated methyl orange dye is given in Figure 9a. The presence of a peak at 3445 cm$^{-1}$ indicates the presence of amine N–H stretching. The N=N stretching shown can be confirmed at a peak around 1603 cm$^{-1}$. The aromatic -C=C stretch is observable at 1422 cm$^{-1}$. The benzene ring connected N–H group peaks are there on range from 815 to 749 cm$^{-1}$, whereas the C–H stretches are in the range 699–900 cm$^{-1}$.

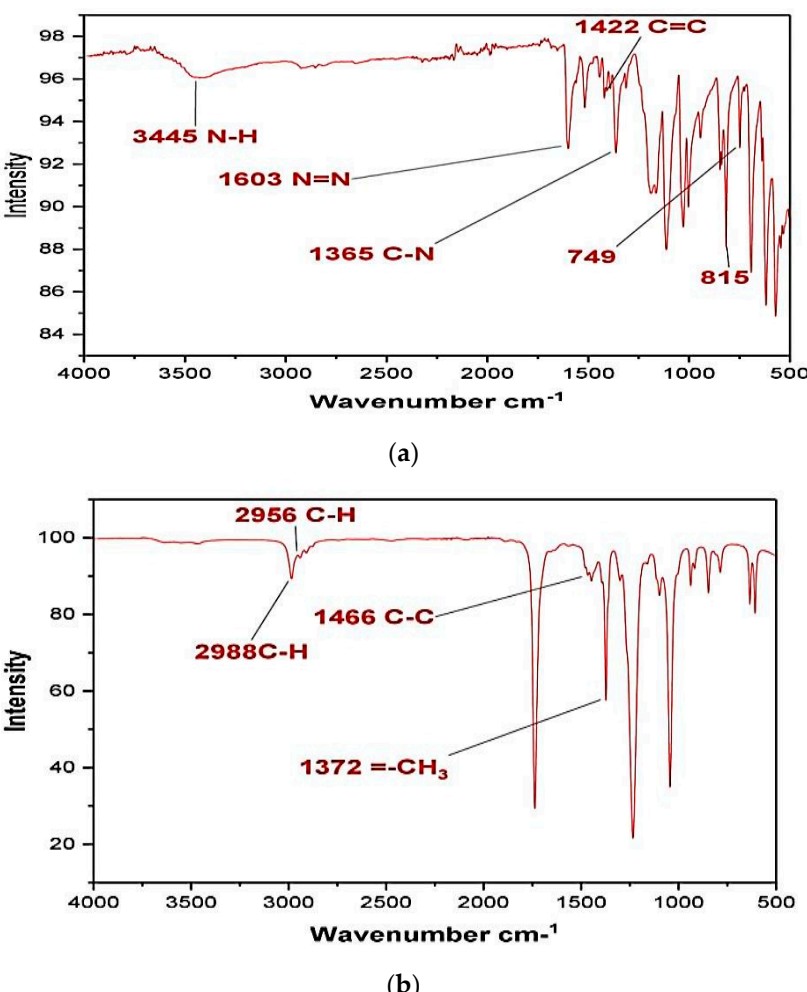

**Figure 9.** (**a**) Methyl orange original dye FTIR. (**b**) Methyl orange dye FTIR after *Pseudomonas aeruginosa* degradation.

The FTIR analyses were also performed for the supernatants collected after application of selected bacteria which in fact were a mixture of undegraded dye and impurities in the solvent. Although significant changes have occurred in the spectrum from that of the original dye spectrum, we are not certain about the products formed or whether these originated from the solvent. Figure 9b shows the FTIR spectrum of dye after *Pseudomonas aeruginosa* degradation. Some peaks of original MO have disappeared and some new peaks have occurred showing drastic changes in the dye structure. The peak at 1603 cm$^{-1}$ has disappeared, pointing towards the breaking of the azo linkage by bacterial azoreductase that is a common enzyme in most bacterial strains [28]. The peak at 2988 cm$^{-1}$ corresponds to the stretching of the =C-H bond on the benzene ring, while the C-H stretching of the methyl group is responsible for the peak at 2956 cm$^{-1}$. The peak at 1466 cm$^{-1}$ reflects C-C bond stretching. The CH$_3$ stretch is evident from the peak at 1372 cm$^{-1}$. The two spectra in Figure 9a,b are distinctively different from each other thus making the correlation extremely difficult. However, the absence of the azo bond peak in bacterially degraded dye points towards degradation rather than decolorization.

*3.10. Gas Chromatography and Mass Spectrometry*

Figure 10 shows the GC-MS chromatograms of the supernatants are a mixture of solvent impurity and degraded products of dye either from bacterial action or hot evaporation in GC analysis. Table 1 shows compounds which are identified from the chromatograms as shown in Figure 11a–d. The detected compounds are most probably originating from the solvent and might be there as an impurity as their presence was not confirmed by

carbon-13 and proton NMR. This means that the complete mineralization of methyl orange has occurred under bacterial degradation by *Pseudomonas aeruginosa*.

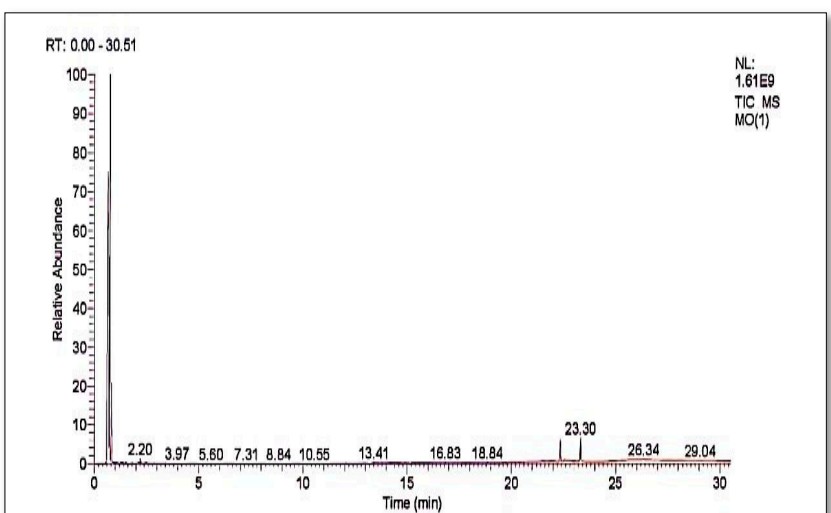

**Figure 10.** GC chromatogram of methyl orange dye after *Pseudomonas aeruginosa* degradation.

**Table 1.** GC-MS analysis and Identified compounds from degraded mixture.

| S.No. | Name of Compound | Peak Area | Retention Time | Molecular Weight | Chemical Formula |
|---|---|---|---|---|---|
| 1 | Toluene | 0.32 | 1.30 | 92 | $C_7H_8$ |
| 2 | p-Xylene | 0.39 | 2.20 | 106 | $C_8H_{10}$ |
| 3 | Cyclohexane, propyl | 0.02 | 2.91 | 126 | $C_9H_{18}$ |
| 4 | 1,2-Benzenedicarboxylic acid, diisooctyl ester | 3.21 | 22.33 | 390 | $C_{24}H_{38}O_4$ |

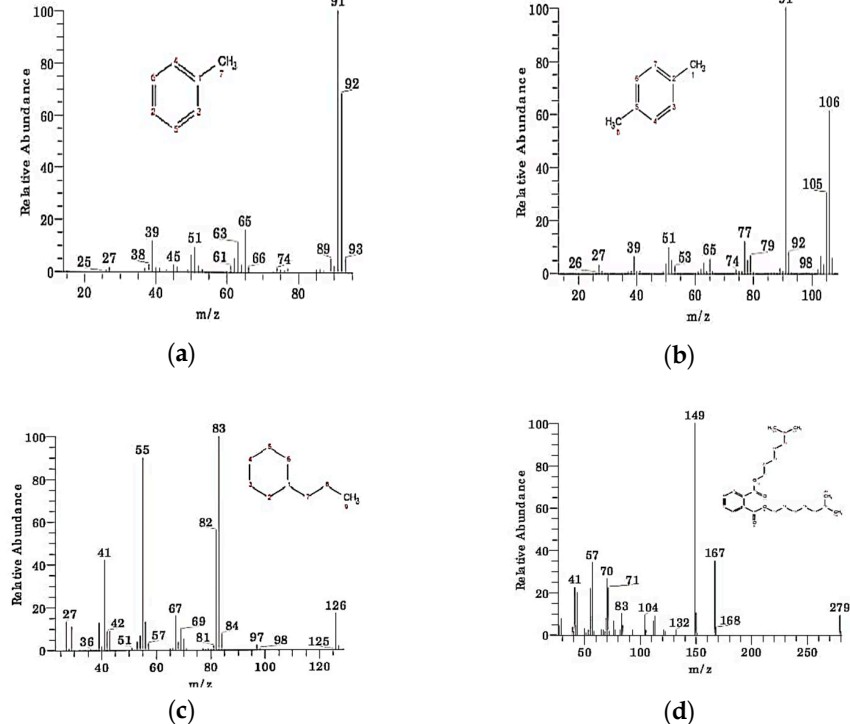

**Figure 11.** GC-MS chromatograms of mixtures of metabolites (**a**) Toluene (**b**) p-Xylene (**c**) Cyclohexane propyl (**d**) 1,2-Benzenedicarboxylic acid, diisooctyl ester.

### 3.11. NMR Spectra of the Dye

When the original dye methyl orange $^1$H NMR spectrum was matched to the spectra generated after bacterial strain biodegradation, a significant variation was observed. Figure 12a,b and Figure 13a,b shows the $^1$H NMR and $^{13}$C NMR of original MO and the column effluent spectrum. The original dye NMR data matched with reported spectral data of MO in the literature [28] whereas the NMR spectra of degraded products did not match with compounds identified in GC-MS analysis or any part of the dye structure after breakage of the azo linkage.

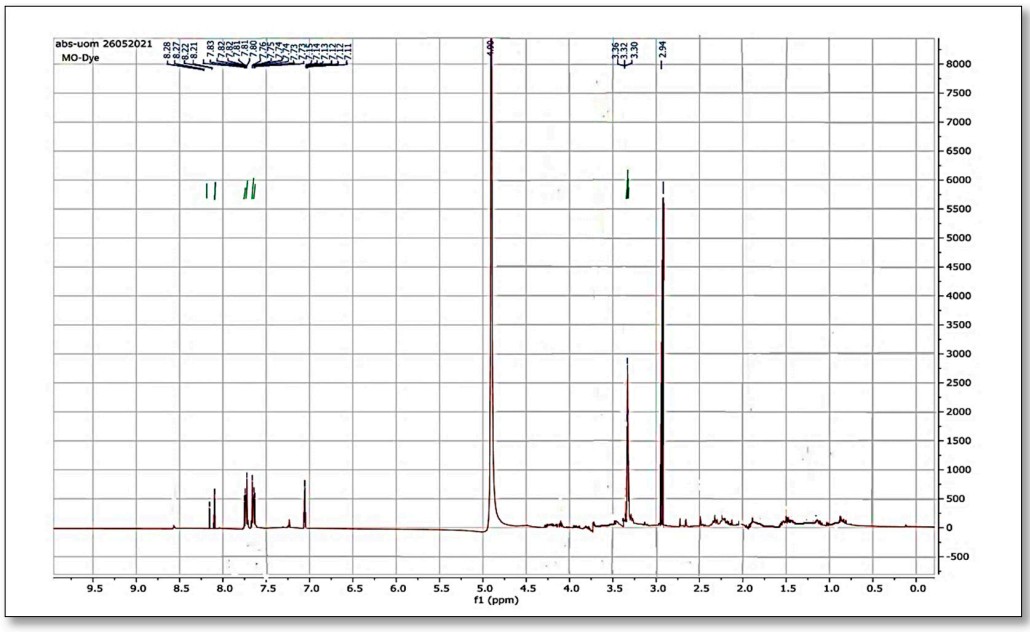

(**a**)

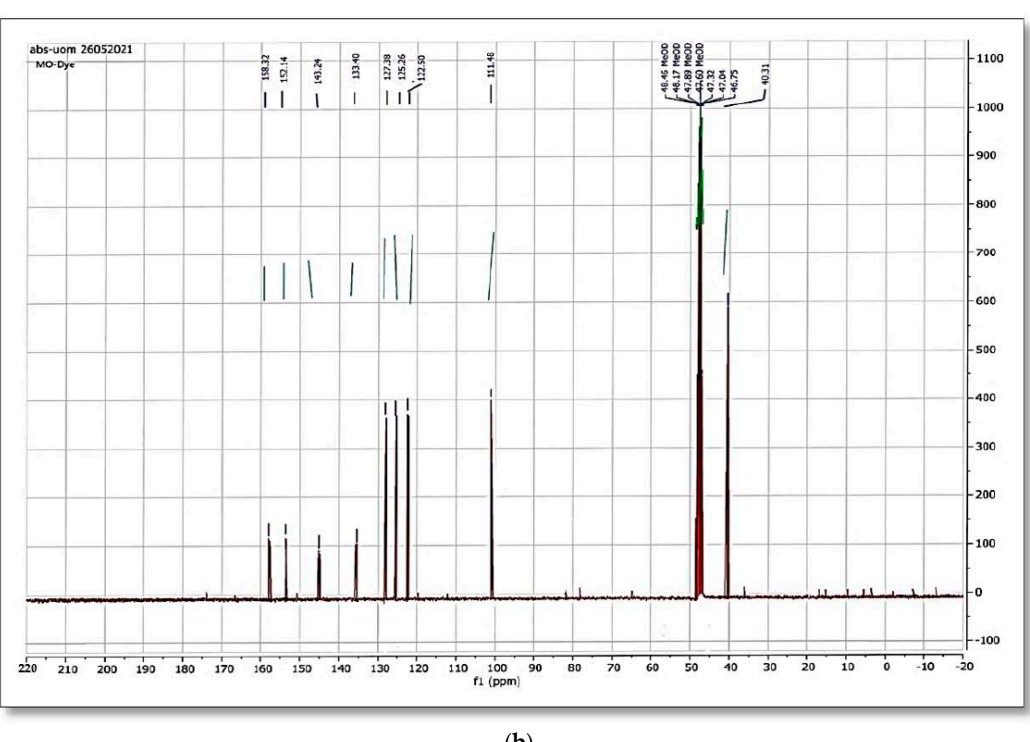

(**b**)

**Figure 12.** (**a**) Original dye methyl orange $^1$H NMR. (**b**) $^{13}$C NMR spectroscopy of the original dye, methyl orange.

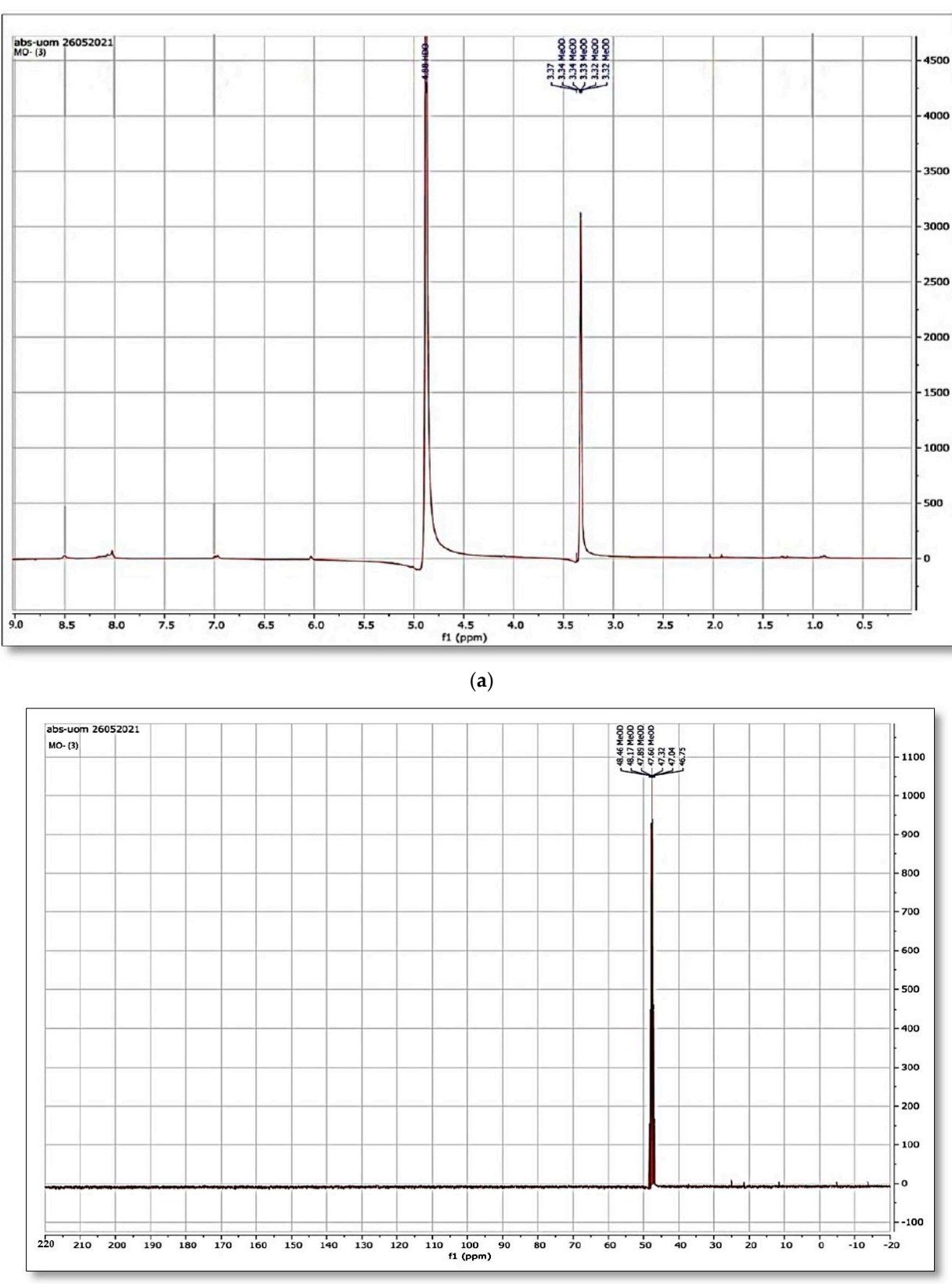

**Figure 13.** (**a**) 1H NMR degradation by *Pseudomonas aeruginosa*. (**b**) $^{13}$C NMR after *Pseudomonas aeruginosa* methyl orange degradation.

1H NMR spectra: The multiple peaks at 3.32, 3.33, 3.34, 3.37, and 4.88 δppm were assigned as methanol. $^{13}$C NMR of methyl orange showed the peaks at 46.75, 47.04, 47.32, 47.60, 47.89, 48.17 and 48.46 were assigned as methanol carbons.

Only solvent peaks are prominent with some impurity peaks that might originate from solvents used in extraction of metabolites and column effluents thus pointing towards the complete mineralization of MO as reported previously.

### 3.12. Proposed Mechanism of Biodegradation

The enzymes azoreductase, peroxidase and laccase are found in bacteria, and these enzymes are responsible for the degradation of azo linkages in the dye structure. Under both anaerobic and aerobic circumstances, the azoreductase enzyme is able to breakdown the azo dye thus splitting the -N=N- bond. The two substituted benzene derivatives are generated as a result of breakage. Under anaerobic environments in capped tubes, the benzene derivatives were completely degraded by bacterial enzymes such as laccases, peroxidases, and polyphenol oxidases. As a result, complete mineralization of methyl orange has occurred. The suggested method of methyl orange breakdown by *Pseudomonas aeruginosa* is shown in Scheme 1. Although Kishor et al. [29] have also reported the complete mineralization of MO using *Pseudomonas aeruginosa*, they have not proposed its stepwise mechanism and also have not provided any spectral information. After evaluating the spectral data we also reached a conclusion that complete mineralization of MO has occurred. Although it is still too early to generate a clear picture of the process, a simplified schematic mechanism has been proposed as shown below in Scheme 1, which needs to be proved by trapping the individual step metabolites using suitable inhibitors of the responsible enzymes.

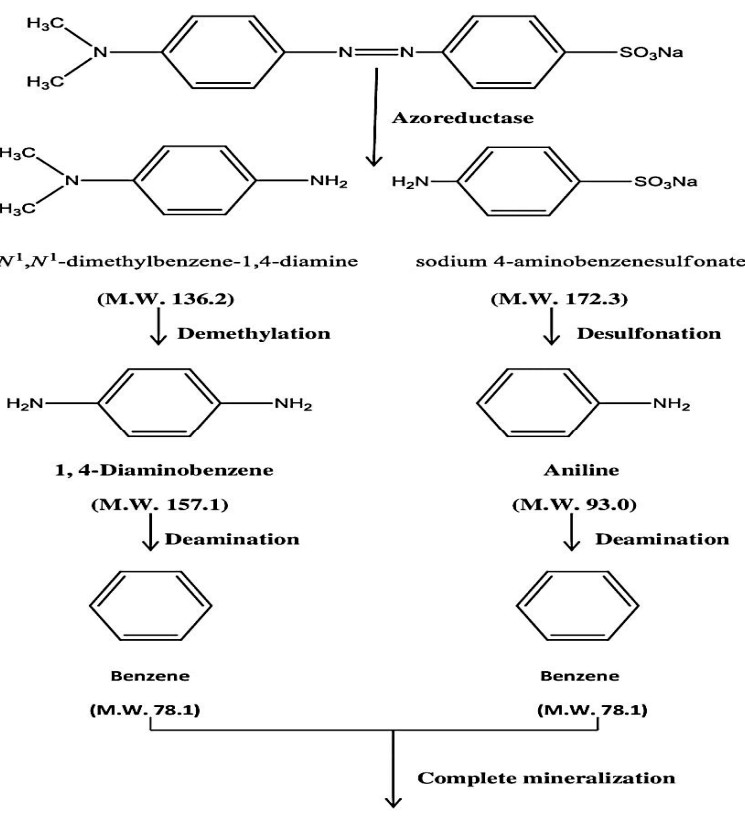

**Scheme 1.** The proposed mechanism of degradation through which *Pseudomonas aeruginosa* degrades methyl orange.

## 4. Conclusions

In this study an attempt was made to degrade MO using *Pseudomonas aeruginosa*. The optimum degradation took place over a three-day time interval, pH 7, 0.5 g of glucose supplementation, 20 ppm dye concentration, 37 °C temperature, and 0.1 g of NaCl tolerable salt concentration. About 88.23% degradation was achieved when all optimal conditions were combined in a single experiment. The products of the final experiment were analyzed using GC-MS, FTIR spectroscopy, whereas NMR analysis were performed for column isolates. The NMR data showed no peaks corresponding to the compounds detected by GC-MS analysis or peaks corresponding to fragments of the dye resulting from azo bond breaking. This indicated complete mineralization of the selected dye. Further experiments are needed to validate the proposed mechanisms of degradation.

**Author Contributions:** Conceptualization, A.U.K. and M.Z.; methodology, A.U.K., A.B.S. and M.Z.; software, A.U.K., A.B.S. and M.Z.; validation, A.U.K.; formal analysis, F.A.K. and I.Z.; investigation, I.Z.; resources, M.Z., G.M.A., R.U., R.B. and H.R.H.M.; data curation; G.M.A., R.U., R.B. and H.R.H.M.; writing original draft preparation: A.U.K. and M.Z.; project administration: M.Z., M.U.R., G.M.A., R.U., R.B. and H.R.H.M. All authors have read and agreed to the published version of the manuscript.

**Funding:** This research work was funded by Princess Nourah bint Abdulrahman University researchers Supporting Project number (PNURSP-2022R30), by Princess Nourah bint Abdulrahman University, Riyadh, Saudi Arabia.

**Institutional Review Board Statement:** Not applicable.

**Informed Consent Statement:** Not applicable.

**Data Availability Statement:** The date presented in this study are available on request from the corresponding author.

**Acknowledgments:** This research work was supported by Princess Nourah bint Abdulrahman University researchers Supporting Project number (PNURSP-2022R30), by Princess Nourah bint Abdulrahman University, Riyadh, Saudi Arabia.

**Conflicts of Interest:** The authors declare no conflict of interest. The funders had no role in the design of the study; in the collection, analyses or interpretation of data; in the writing of the manuscript or in the decision to publish the results.

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
