# Peer review of "Biological Mineralization of Methyl Orange by Pseudomonas aeruginosa"

_water, doi:10.3390/w14101551_

Round 1
Reviewer 1 Report
- The work is very valuable for the scientific community and for the protection of the environment. So far none of the researchers have given attention to the nature of metabolites formed. Such types of studies are important from environmental point of view to know whether the resulting metabolites are toxic or eco-friendly;
- However, on lines 82-84 it says: Yang et al., [12] reported that Stenotrophomonas acidaminiphila degrades the dye methyl orange up to 95% at optimum conditions. When writing about the optimal conditions, either quote them or omit this provision;
- Please enter the manufacturer MB. This is an analogy to other reagents.
- Where did the bacterial inoculum come from? (the line 116);
- Why MB degradation was only done for 3 days. What are the reasons for this. What are the OECD recommendations? Please link this to chapter 2.3.6;
- Please correct title 3.1, caption under Figure 4, descriptions of axes in the diagrams;
- Figures 12 a and b and 13 a and b are completely illegible;
- In my opinion, further experiments are needed to verify the proposed mechanism, special attention should be paid in time. I think it's for the next research and publications;
Author Response
Reviewer 1:
- The work is very valuable for the scientific community and for the protection of the environment. So far none of the researchers have given attention to the nature of metabolites formed. Such types of studies are important from environmental point of view to know whether the resulting metabolites are toxic or eco-friendly;
- Worthy reviewer, thank you for the encouraging remarks. According to the NMR data, there are no metabolites identified that indicate that the entire mineralization of methyl orange has taken place.
- However, on lines 82-84 it says: Yang et al., [12] reported that Stenotrophomonas acidaminiphila degrades the dye methyl orange up to 95% at optimum conditions. When writing about the optimal conditions either quote them or omit this provision;
- Worthy reviewer, Only the percentage of degradation is included, while quoting the optimum conditions whereas in our study first we have optimized the conditions singly and then combined all optimized conditions where an 23% degradation have been observed.
- Please enter the manufacturer MB. This is an analogy to other reagents.
- Worthy reviewer, mistakenly it was written as MB which is now corrected as MO. The manufacturer details were incorporated accordingly in the revised manuscript.
- Where did the bacterial inoculum come from? (the line 116);
- Worthy reviewer, they have been identified and isolated from industrial contaminated waters. The required detail were incorporated in the revised paper accordingly.
- Why MB degradation was only done for 3 days. What are the reasons for this. What are the OECD recommendations? Please link this to chapter 2.3.6;
- Worthy reviewer, three days was decided after execution of optimum time experiments where maximum degradation was observed after 3 days. Figure 8 shows the results of that experiment. No substantial degradation was seen after three days. According to OECD recommendations, the standardized time for degradation permissible is 28 days whereas P. aeruginosa is capable of growing in conditions of extremely low nutrient content. This species can survive and proliferate in water for up to 100 days or longer. So worthy reviewer, 3 days’ time interval falls within this limit. The required detail has been incorporated in section 2.3.6 accordingly.
- Please correct title 3.1, caption under Figure 4, descriptions of axes in the diagrams;
- Worthy reviewer, the corrections required were made accordingly.
- Figures 12 a and b and 13 a and b are completely illegible;
- Worthy reviewer: The size of the figure along with resolution was increased in order to improve the overall quality of the figures.
- In my opinion, further experiments are needed to verify the proposed mechanism, special attention should be paid in time. I think it's for the next research and publications;
- Worthy reviewer, we know at this stage it is too early to make such claims. Further validation of the mechanism is important. Our work is in progress and hopefully we will incorporate the valuable suggestion of the worthy reviewer in our next studies.
Reviewer 2 Report
The manuscript deals with the "decolorization of MO by Pseudomonas aeruginosa".
In recent years, the unprecedented development of industrial and urban activities has led to a significant increase in wastewater discharge into the environment, often contaminated with harmful organic pollutants (e.g., dyestuffs). Therefore, the separation/elimination of these pollutants from water sources is a goal that must be accomplished to ensure human and environmental safety. Due to their high toxicity, chemical stability, and low biodegradability, dyes are a class of pollutants that are raising increasing concern, since they cause severe problems to aquatic life and human beings. Application of biological methods in reducing the effects of dyes is one of the promising techniques.
1. Decolorization of dyes from aqueous solutions by Pseudomonas has been widely reported; therefore, what is the novelty of your study?
https://doi.org/10.1007/s13205-013-0192-7
https://doi.org/10.1016/j.jwpe.2021.102300
https://doi.org/10.1016/j.jenvman.2020.110383
2. Page 1, Line 17; "Due to the recalcitrant and carcinogenic nature, the presence of methyl orange (MB) in ..." MB shows the methyl blue; thus, please use MO for abbreviation of methyl orange!
3. Page 1, Line 26; "was achieved at 37°C, a pH of 7, low salt concentration, a high carbon source,..." Mention the value of "low salt concentration" and high carbon source".
4. By reading the Introduction, I could not find any gap of the knowledge which encourages you to use Pseudomonas for decolorization of MO! Why did you use this bacteria sp.? Never used before?! What makes your study unique?!
5. The quality of Fig. 12 and 13 should be improved.
Author Response
Reviewer 2:
The manuscript deals with the "decolorization of MO by Pseudomonas aeruginosa".
In recent years, the unprecedented development of industrial and urban activities has led to a significant increase in wastewater discharge into the environment, often contaminated with harmful organic pollutants (e.g., dyestuffs). Therefore, the separation/elimination of these pollutants from water sources is a goal that must be accomplished to ensure human and environmental safety. Due to their high toxicity, chemical stability, and low biodegradability, dyes are a class of pollutants that are raising increasing concern, since they cause severe problems to aquatic life and human beings. Application of biological methods in reducing the effects of dyes is one of the promising techniques.
- Decolorization of dyes from aqueous solutions by Pseudomonas has been widely reported; therefore, what is the novelty of your study?
https://doi.org/10.1007/s13205-013-0192-7
https://doi.org/10.1016/j.jwpe.2021.102300
https://doi.org/10.1016/j.jenvman.2020.110383
- Worthy reviewer: Pseudomonas aeruginosa is a facultative bacterium that can grow both aerobically and anaerobically, and has the ability to rapidly grow in azo dye contaminated water. For the characterization of methyl orange metabolites, no study has been conducted before this research. Also worthy reviewer, the mentioned studies are not dealing with the nature and identification of the metabolites. Here we have not only isolated them but have proposed a mechanism of the selected dye degradation by bacteria as well.
- Page 1, Line 17; "Due to the recalcitrant and carcinogenic nature, the presence of methyl orange (MB)in ..." MB shows the methyl blue; thus, please use MO for abbreviation of methyl orange!
- Worthy reviewer, thank you so much for the correction, it was corrected as MO (Methyl Orange) instead of MB in the revised paper.
- Page 1, Line 26; "was achieved at 37°C, a pH of 7, low salt concentration, a high carbon source,..." Mention the value of "low salt concentration" and high carbon source".
- Worthy reviewer, thank you for the valuable suggestion; corrected as, low salt concentration of 0.1 g/15ml and a high carbon source of 0.6 g/15ml.
- By reading the Introduction, I could not find any gap of the knowledge which encourages you to use
Pseudomonas for decolorization of MO! Why did you use this bacteria sp.? Never used before?! What makes your study unique?!
- Worthy reviewer: “Pseudomonas aeruginosa is a facultative bacteria that can grow both aerobically and anaerobically, and has the ability to rapidly grow in azo dye contaminated water. For the characterization of methyl orange metabolites, no study has been conducted before this research.” This statement has been added accordingly to cover the gaps. Hope this will be ok
- The quality of Fig. 12 and 13 should be improved.
- Worthy reviewer: The size and resolution of the these figures were increased in order to improve their overall quality.
Round 2
Reviewer 2 Report
Reviewers' comments have been addressed.
This manuscript is a resubmission of an earlier submission. The following is a list of the peer review reports and author responses from that submission.